# Geochemical Evidence of Ediacaran Phosphate Nodules in the Volyno-Podillya-Moldavia Basin, Ukraine

Ahmet Sasmaz [1,*] , Bilge Sasmaz [1], Yevheniia Soldatenko [2], Abderrazak El Albani [3], Edward Zhovinsky [4] and Nataliya Kryuchenko [4]

1   Department of Geological Engineering, Firat University, 23119 Elazig, Turkey
2   Department of General and Structural Geology, Dnipro University of Technology, 49005 Dnipro, Ukraine
3   Université de Poitiers, L'IC2MP (Institut de Chimie des Milieux et Matériaux de Poitiers),
    B35, TSA 51106, 5 Rue Albert Turpain, 86073 Poitiers, France
4   Mineralogy and Ore Formation of National Academy of Science,
    M.P. Semenenko Institute of Geochemistry, 03680 Kiev, Ukraine
*   Correspondence: asasmaz@firat.edu.tr

**Abstract:** The sedimentary basin of Podillya (Volyno-Podillya-Moldavia) is situated in the southwest of the Ukrainian crystalline shield and belongs to the middle part of the Upper Neoproterozoic section of the Moguiliv-Podilska Group. By analyzing the primary oxide, trace, and rare-earth element compositions of the phosphate nodules in the area, this study sought to shed light on the potential precipitation characteristics of the Ediacaran Sea, where phosphate nodules were created. The mean major oxide contents of the nodules were 50.8 wt.% CaO, 34.2 wt.% $P_2O_5$, 5.29 wt.% $SiO_2$, 4.77 wt% LOI, 1.69 wt% $Fe_2O_3$, 1.63 wt% $Al_2O_3$, and 0.35 wt.% MnO. The average trace element concentrations were 183 ppm Ba, 395 ppm Sr, 13.4 ppm Ni, 32.7 ppm Cr, 62.2 ppm Zn, 764 ppm Y, 16 ppm V, 10.8 ppm As, 75.8 ppm Cu, 84 ppm Pb, 2.1 ppm U, 1.7 ppm Th, and 4.2 ppm Co. The trace element contents were generally low and indicated an assemblage of Cu, Y, As, Cd, and Pb enrichments in comparison to PAAS. The total REE concentrations varied from 1638 ppm to 3602 ppm. The nodules had medium REE (MREE) enrichments and showed similar REE patterns normalized to PAAS. All the nodules had strongly negative Ce, Pr, and Y anomalies and substantially negative Eu anomalies, with four samples being exceptions. These abnormalities suggest that oxic and suboxic sea conditions existed at the time the nodules formed. The extremely high REE concentrations are thought to be the result of REEs being redistributed between the authigenic and detrital phases that were created during the diagenetic equilibration of phosphate with pore water. The genetic hypothesis for phosphate nodule formation states that the nodules were generally formed in oxic and suboxic seawater and were precipitated on slopes in response to a significant upwelling from a deeper basin with abundant organic matter under anoxic/suboxic conditions. The majority of the organic material at the water–sediment interface of the seafloor underwent oxidation before phosphate was released into the pore water of the sediment.

**Keywords:** phosphate; nodule; ediacaran; REE geochemistry; Podillya Basin; Ukraine

## 1. Introduction

Over the last 3.85 billion years of Earth's history, various geological events have occurred in distinct periods. Notably, atmospheric oxygen levels rose to values higher than 0.2 atm during the Proterozoic Eon, specifically between 0.80 and 0.54 billion years ago. This increase in oxygen was accompanied by a similar trend in the shallow oceans [1]. The phosphate deposits between 0.8 Ga and 0.54 Ga began to create sedimentary records such as sedimentary manganese deposits and banded iron formations [2]. Modern and historic sea sediments contain a variety of phosphate rocks. The creation of phosphate is known to occur now in continental margins and continental shelves in upwelling places such as the Gulf of California, Namibia, Chile, and Peru [3–6]. Phosphorite accumulations

formed during the Ediacaran–Cambrian transition [7]. Phosphorites formed in this time period have been well studied in many paleogeographic areas, including Australia [8], China [9–11], the West African craton [12–14], India [15,16], the Siberian Platform [17,18], and Mongolia [19,20]. Little research has been conducted on the phosphate accumulations, both economic and subeconomic, that occurred at the western edges of Gondwana and Baltica [21,22].

The Ediacaran biota contains the first complex macroscopic organisms observed in the geological record, overshadowing the radiation of eumetazoan animals during the Cambrian explosion. However, there is little information about the quality of food sources and the possible roles of nutrient availability [23]. Moreover, marine sedimentary rocks in the middle-to-late Ediacaran (575–541 Ma) contain the first samples of macroscopic, multicellular-bodied fossils, but likewise, little information about their food sources and environments exists [24]. As has been reported by chemostratigraphic analyses, the Ediacaran oceans changed from predominantly oxic (well ventilated) to more thoroughly oxygenated [25,26]. Trace fossils also show a marked increase in diversity during this time interval (Middle and Late Ediacaran (ca. 575–541 Ma)) [27,28]. Oxygen has been offered as a factor for the evolution and rise of metazoans and multicellular biota. Isotopic records and trace element changes show the ocean surface oxygenation event at about 850 Ma [25,29,30]. Although anoxic and even euxinic marine environments were defined in the Ediacaran period, the redox conditions of the environments containing Ediacaran biota indicate oxic conditions [31–33].

The abundance of REEs, as well as Ce and Eu anomalies, in the phosphate samples in different regions has been investigated by various researchers [34–43]. Due to the redox conditions and composition of the depositional environment, phosphates typically contain a lot of REEs and exhibit negative Ce anomalies [44–46]. The Volyno-Podillya-Moldavia Basin is a very interesting region for investigating the redox conditions and composition of the depositional environment. Whereas there have been many studies about the mineralogy, geology, and formation of phosphate nodules in the Volyno-Podillya-Moldavia Basin [47–53], there has not been a study focused on the trace element and REE geochemistry. The objectives of this research are to examine the major oxide, trace, and rare-earth element concentrations of the phosphate nodules under study, as well as to identify the physicochemical properties of the paleoenvironment in which they were deposited.

## 2. Geological Setting

The Volyno-Podillya-Moldavia Basin is located in the southwest of Ukraine at the edge of the Ukrainian crystalline shield (Figures 1 and 2) and is one of a complex network of Neoproterozoic and Paleozoic sedimentary basins that developed on the western edge of the East European Craton. The host rocks containing phosphates in Transnistria are represented by siltstones of greenish gray, blue-gray, and brown colors, as well as typical dark-brown and black clayey shales with thin layers of argillites of dark-gray, light-green, greenish-gray, and brown colors. Phosphate nodules are always confined to the upper part of the horizon of dark-gray argillites, except for the Chocin area. From the northeast to the southwest, the sedimentary strata of Ediacaran time overlie a granitic basement complex rock; sedimentary rocks lie at a low angle. Above, they are covered by younger Paleozoic rocks; they outcrop to the western part of the basin (Figure 2). This sedimentary basin is constrained to the Ediacaran Period by biostratigraphical and geochronological proxies [54]. The Neoproterozoic Podillya Sedimentary Basin is known to host imprints of Ediacaran soft-bodied fauna [50–57]. From this period, fossil-rich silisiclastic sediments, recognized as traces of early metazoans, also contain evidence of significant microbiological activity. Many structures in these Ediacaran sediments can be interpreted as microbial-induced sedimentary structures [55,57].

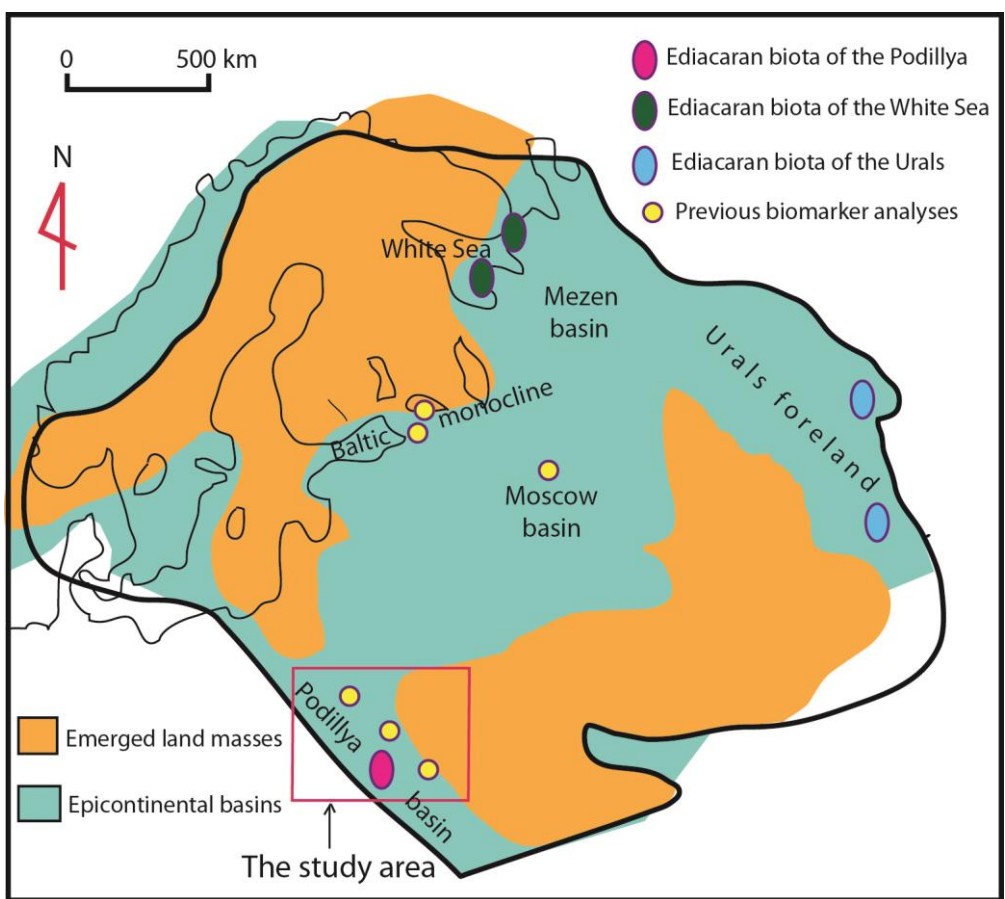

**Figure 1.** Paleogeography and localities of Ediacaran phosphate deposits in the East European Platform (modified after [24,58] and Baltica at ~550 Ma (modified from [59]).

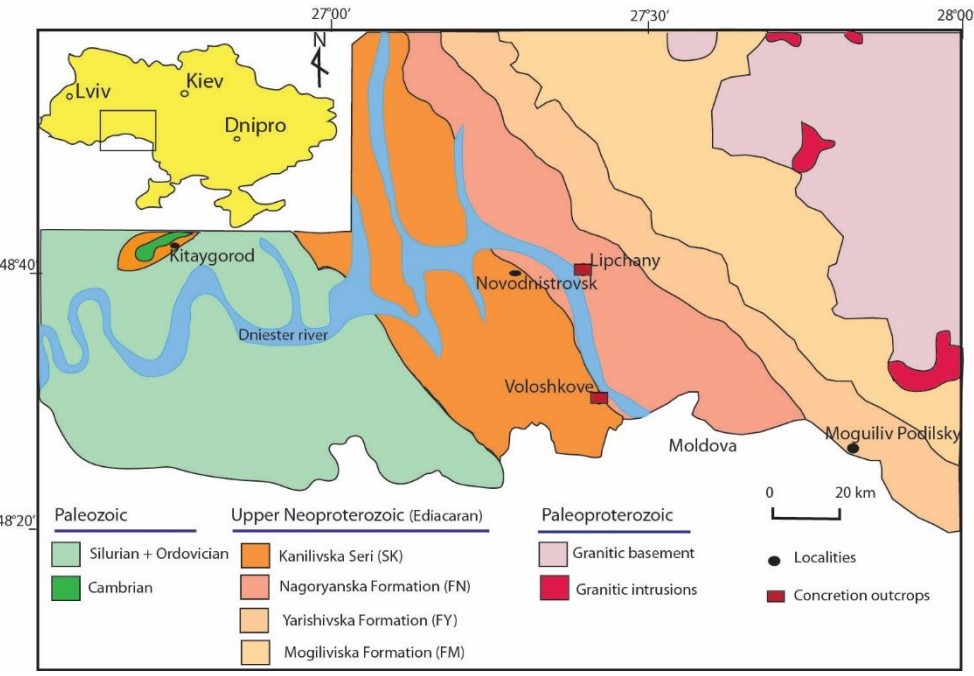

**Figure 2.** Geological map of the study area and sample locations of phosphate nodules outcrops. Data from [51].

The outcrops from which phosphate nodule samples were taken for this study are located along the Dniester basin. The main two locations for sampling were around the villages of Lipchany and Voloshkove (Figure 2). Stratigraphically, the phosphate nodules belong to the middle part of the upper Neoproterozoic section of the Moguiliv-Podilska Group, Nagoryanska Formation, Kalus Sequence (Figure 3). This section is characterized by either clayey sand or sandy units, with an absolute dominance of clayey, thin-layered, dark-colored, sometimes black shales, according to several studies [50,56]. Phosphate nodules close to the village of the Voloshkove are located on the border between Ukraine and Moldova. These layers, containing dense phosphate nodules, are composed of fine, clayey and sandy, nonrhythmic levels varying from greenish brown to dark brown at a distance of less than one meter (Figure 4a). These nodules are arranged regularly between the layers with a diameter of 8–10 cm (Figure 4b,c). Phosphate nodules have been observed in the argillitic facies, where they form alignments parallel to the stratification of the host sediment. The whole sequence consists of arrhythmic alternation of fine argillitic material, more or less massive, of general greenish brown color (Figure 4a). The nodules begin to grow around a light-colored phosphate-rich unit in the center, then grow outward as spindle-shaped rods, surrounded by thin pelitic material, as darker phosphates (Figure 4b,c).

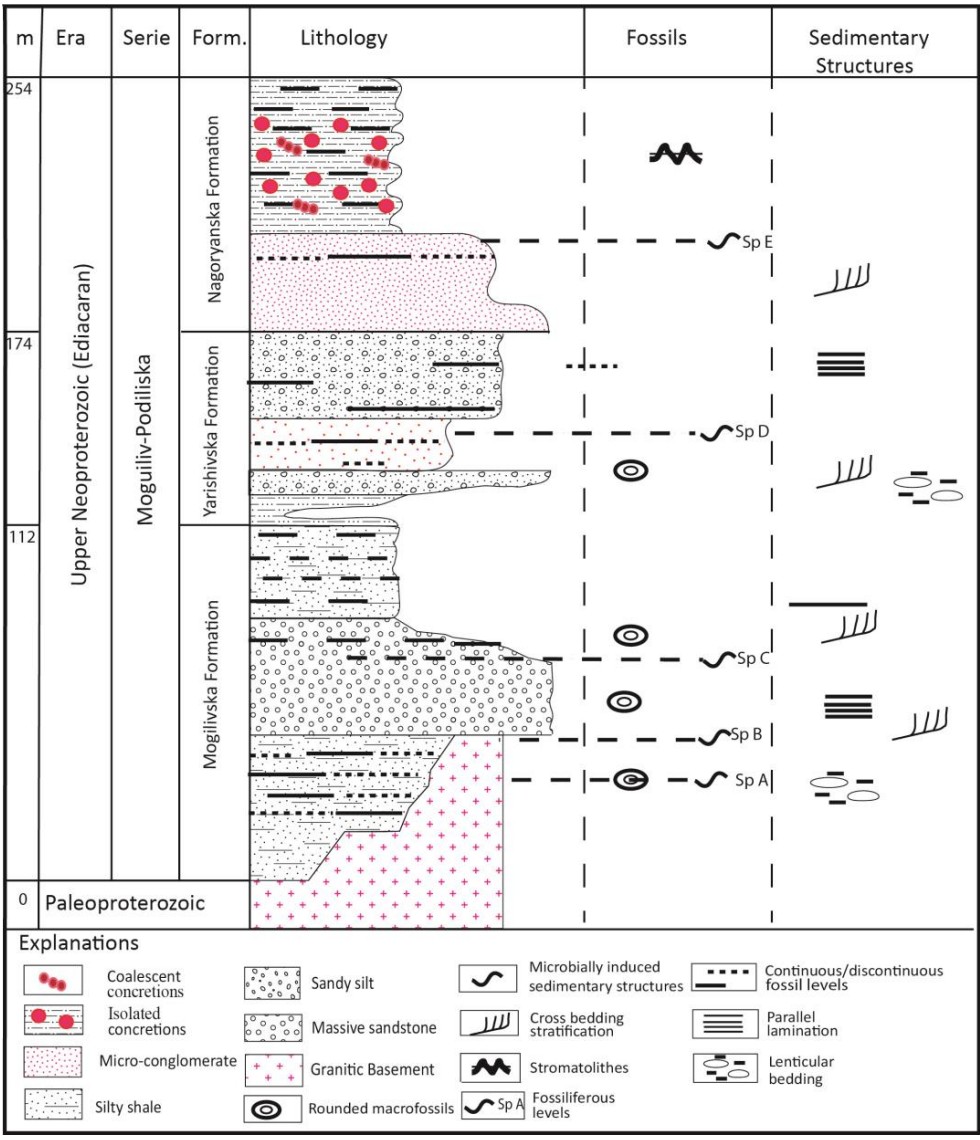

**Figure 3.** Stratigraphy of the Volyno-Podillya-Moldavia Basin, with lithostratigraphic section of studied area (after [47]).

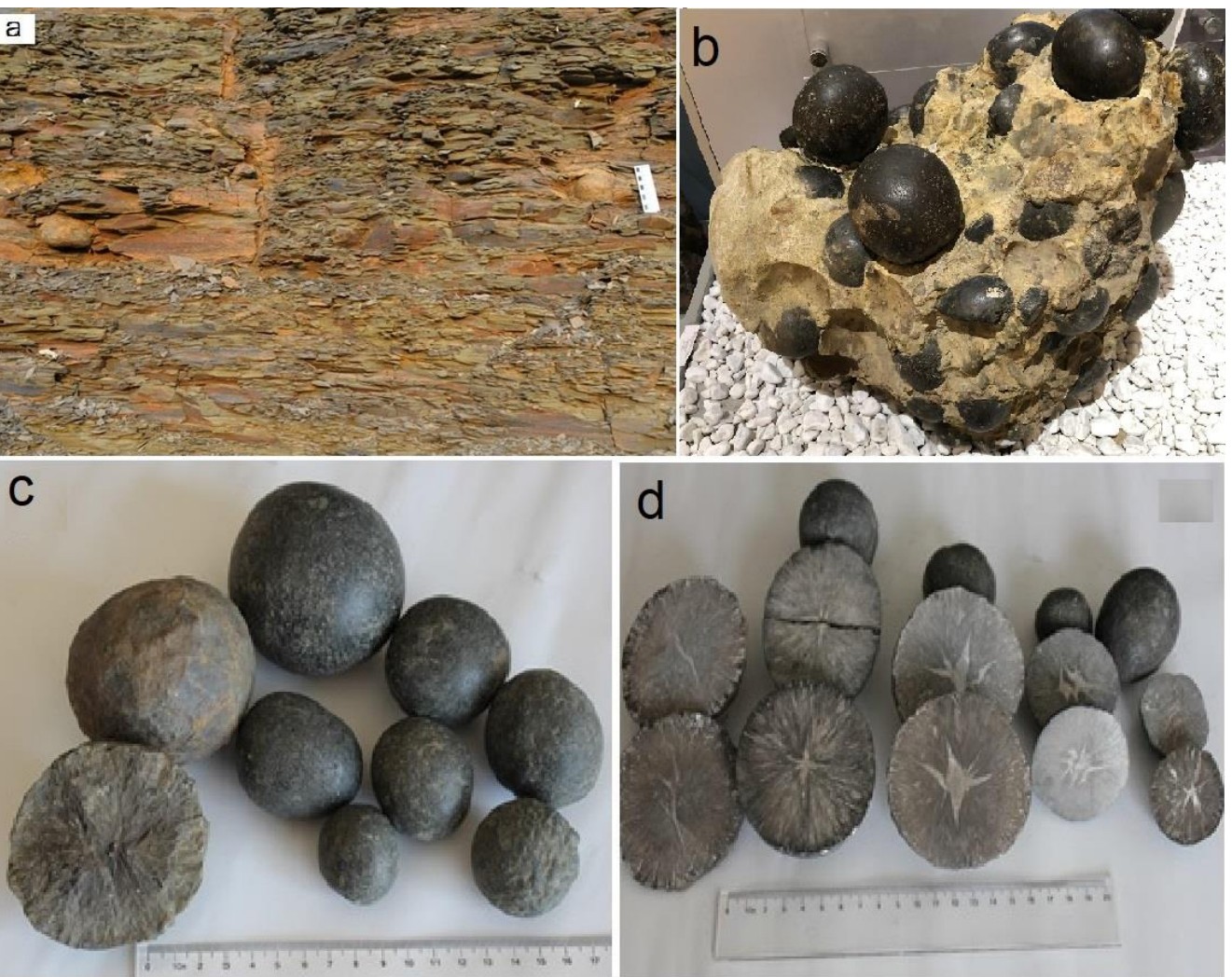

**Figure 4.** (**a**) Distribution of nodules between layers; (**b**–**d**) Phosphate nodules in different dimensions and their internal structures.

## 3. Samples and Analytical Method

### 3.1. Samples

The phosphorite-bearing strata have a width of 15–20 km in the southern region and may continue into Moldovan territory, depending on the geological formations and their distribution in the area. There are very interesting relationships between phosphate nodules and their argillaceous slates. The nodules are typically surrounded by clay shale, forming an eye-like structure. These nodules have diameters ranging from 2–5 cm to 25 cm and can weigh between 0.05 and 15 kg, as shown in Figure 4a. Their outer surface is mostly spherical, often brown, resembling a rusty iron core. They exhibit a mostly rounded shape, with some being flattened or arranged in layers that have grown together, forming concentric rings that expand outward from an independent center. In many instances, smaller balls are found among the larger ones, forming a cluster-like structure (Figure 4b–d). Thirteen representative phosphate nodule samples were collected from the clay layers in the Neoproterozoic Podillya Sedimentary Basin for the analysis of major oxides, trace elements, and rare-earth elements (REEs). The samples, which varied in size, were divided into two sections: one was used for element analysis, while the other was used for mineralogical and petrographic investigations.

### 3.2. Analytical Method

The phosphate nodules were cut and crushed to a 200-mesh size and then analyzed for rare-earth elements, trace elements, and major oxides. Analytical uncertainties were observed within 5% for each element. Using inductively coupled plasma atomic emission spectroscopy, the main oxides were examined, while rare-earth and trace element concentrations were examined using inductively coupled plasma mass spectrometry. All analyses were carried out at the Bureau Veritas Minerals (BVM) Laboratory (commercial laboratory) in Vancouver, Canada. Powdered phosphate nodules were dissolved by mixing $HNO_3$:HCl:$H_2O$ (1:1:1, *v/v*; 6 mL per 1.0 g of phosphate samples) for one hour.

The analytical data were analyzed statistically using the Student–Newman–Keuls Procedure (SNK) with SPSS 15.0 software (IBM Corp., Armonk, NY, USA) and ANOVA (analysis of variance).

The Ce, Eu, and Y anomaly values of the phosphate nodules were calculated according to Taylor and McLennan's [60] PAAS values using the following formulas: $Y/Y* = Y_n/\sqrt{[Dy_n * Ho_n]}$, $Eu/Eu* = Eu_n/\sqrt{[Sm_n * Gd_n]}$ and $Ce/Ce* = Ce_n/\sqrt{[La_n * Pr_n]}$.

### 3.3. Quality Assurance

The Bureau Veritas Minerals (BVM) Analytical Laboratory has ISO registrations and accreditations at all of its sites. These certifications and registrations offer third-party verification and abide by ISO standards. All BVM facilities have ISO 9001 registrations and are awaiting Bureau Veritas corporate registration. The ISO/IEC 17025 accreditation for particular laboratory techniques has also been granted to a number of analytical centers.

## 4. Results and Discussion

### 4.1. Major Oxide Geochemistry

CaO was the most abundant major oxide in the studied samples with an average of $50.8 \pm 2.32$ wt.%. The average $P_2O_5$ concentration was $34.2 \pm 1.82$ wt.%, making it the second most abundant oxide within the studied phosphate samples, with high average contents varying from 29.6 to 38.4 wt.%. The average concentrations of $SiO_2$ and $Fe_2O_3$ were $5.29 \pm 0.23$ wt.% and $1.69 \pm 0.12$ wt.%, respectively (Table 1). The average contents of $Al_2O_3$ and LOI were $1.63 \pm 0.11$ and $4.77 \pm 0.36$ wt%, respectively. The averages of the other major oxides were MnO: $0.35 \pm 0.02$ wt%; $K_2O$: $0.31 \pm 0.02$ wt.%; MgO: $0.24 \pm 0.01$ wt.%; $Na_2O$: $0.20 \pm 0.01$ wt.%; $TiO_2$: $0.04 \pm 0.01$ wt.%; and $Cr_2O_3$: $0.003 \pm 0.01$ wt.%) (Table 1). The major oxide concentrations of the nodules decreased in the following order: CaO > $P_2O_5$ > $SiO_2$ > $Fe_2O_3$ > $Al_2O_3$ > MnO > $K_2O$ > MgO > $Na_2O$ > $TiO_2$ > $Fe_2O_3$. The $CaO/P_2O_5$ ratios varied from 1.36 to 1.70 with a mean of 1.49. This ratio is important for the economic significance of phosphate raw materials [61]. The $CaO/P_2O_5$ ratio in phosphate rocks was higher than 1.31, and this may be related to either the occurrence of calcite or dolomite in whole rocks or the substitution of $PO_4$ by $CO_3$ [62,63]. The analysis showed that $P_2O_5$ was highly positively correlated with $K_2O$ (r = 0.87), MgO (r = 0.68), and Rb (r = 0.62) and had weak positive correlations with Ta (r = 0.48), Hf (r = 0.38), and CaO (r = 0.34). In contrast, $P_2O_5$ exhibited strong negative correlations with MnO (r = −0.73) and $Fe_2O_3$ (r = −0.52) and weak negative correlations with $Al_2O_3$ (r = −0.31) and $Na_2O$ (r = −0.31) (Table 2; Figure 5).

**Table 1.** Major oxide contents (%) of the phosphate nodule samples (DL: detection limit of ICP-MS).

| | SiO₂ | Al₂O₃ | Fe₂O₃ | MgO | CaO | Na₂O | K₂O | TiO₂ | P₂O₅ | MnO | LOI | Sum | CaO/P₂O₅ |
|---|---|---|---|---|---|---|---|---|---|---|---|---|---|
| Det. Limit | 0.01 | 0.01 | 0.04 | 0.01 | 0.01 | 0.01 | 0.01 | 0.01 | 0.01 | 0.01 | −5.1 | 0.01 | |
| STD SO-19 | 60.5 | 13.9 | 7.47 | 2.92 | 5.94 | 4.02 | 1.29 | 0.70 | 0.31 | 0.21 | 1.9 | 99.78 | |
| PH-01 | 6.07 | 2.11 | 1.77 | 0.26 | 50.1 | 0.20 | 0.33 | 0.04 | 35.3 | 0.14 | 3.1 | 99.48 | 1.42 |
| PH-02 | 4.48 | 1.35 | 1.57 | 0.28 | 52.3 | 0.13 | 0.34 | 0.03 | 36.9 | 0.08 | 2.2 | 99.67 | 1.41 |
| PH-03 | 5.80 | 1.43 | 1.75 | 0.22 | 50.3 | 0.27 | 0.31 | 0.03 | 34.0 | 0.32 | 4.9 | 99.41 | 1.47 |

**Table 1.** *Cont.*

|  | SiO$_2$ | Al$_2$O$_3$ | Fe$_2$O$_3$ | MgO | CaO | Na$_2$O | K$_2$O | TiO$_2$ | P$_2$O$_5$ | MnO | LOI | Sum | CaO/P$_2$O$_5$ |
|---|---|---|---|---|---|---|---|---|---|---|---|---|---|
| PH-04 | 5.59 | 1.34 | 1.54 | 0.22 | 51.3 | 0.24 | 0.31 | 0.03 | 36.0 | 0.15 | 2.6 | 99.39 | 1.42 |
| PH-05 | 4.14 | 1.08 | 1.50 | 0.26 | 52.1 | 0.14 | 0.36 | 0.03 | 38.4 | 0.04 | 1.6 | 99.57 | 1.36 |
| PH-06 | 5.72 | 1.91 | 1.61 | 0.28 | 50.8 | 0.17 | 0.37 | 0.05 | 36.6 | 0.04 | 1.9 | 99.52 | 1.39 |
| PH-07 | 4.52 | 0.77 | 1.66 | 0.19 | 51.6 | 0.32 | 0.26 | 0.02 | 32.3 | 0.55 | 7.4 | 99.60 | 1.59 |
| PH-08 | 6.39 | 3.06 | 1.60 | 0.18 | 49.1 | 0.14 | 0.28 | 0.06 | 30.3 | 0.41 | 7.8 | 99.25 | 1.62 |
| PH-09 | 5.30 | 2.43 | 2.22 | 0.22 | 49.6 | 0.16 | 0.19 | 0.04 | 29.6 | 0.55 | 9.1 | 99.41 | 1.67 |
| PH-10 | 5.66 | 1.86 | 1.82 | 0.25 | 50.1 | 0.16 | 0.31 | 0.04 | 34.7 | 0.50 | 4.3 | 99.88 | 1.44 |
| PH-11 | 5.27 | 1.30 | 1.74 | 0.22 | 50.8 | 0.24 | 0.30 | 0.03 | 34.0 | 0.43 | 5.4 | 99.92 | 1.50 |
| PH-12 | 6.13 | 1.84 | 1.68 | 0.35 | 49.7 | 0.15 | 0.39 | 0.05 | 35.6 | 0.62 | 3.2 | 99.92 | 1.40 |
| PH-13 | 3.70 | 0.71 | 1.57 | 0.17 | 52.8 | 0.29 | 0.23 | 0.02 | 31.1 | 0.66 | 8.5 | 99.93 | 1.70 |
| Average | 5.29 | 1.63 | 1.69 | 0.24 | 50.8 | 0.20 | 0.31 | 0.04 | 34.2 | 0.35 | 4.77 | 99.61 | 1.49 |

**Table 2.** Correlation coefficients between P$_2$O$_5$ and other major oxides in the phosphate nodules.

|  | SiO$_2$ | Al$_2$O$_3$ | Fe$_2$O$_3$ | MgO | CaO | Na$_2$O | K$_2$O | TiO$_2$ | MnO | Cr$_2$O$_3$ | V | Co | Ni | Cu | Rb |
|---|---|---|---|---|---|---|---|---|---|---|---|---|---|---|---|
| P$_2$O$_5$ | −0.05 | −0.31 | −0.52 | 0.68 | 0.34 | −0.31 | 0.87 | −0.06 | −0.73 | −0.19 | −0.10 | 0.17 | −0.23 | 0.15 | 0.62 |
|  | Sr | Y | Zr | Nb | Ba | Hf | As | Cd | U | Th | Pb | Zn | Ta | Sc | ΣREE |
| P$_2$O$_5$ | −0.29 | −0.27 | 0.21 | −0.19 | −0.39 | 0.38 | 0.22 | 0.07 | 0.12 | 0.31 | −0.01 | 0.10 | 0.48 | −0.10 | −0.15 |

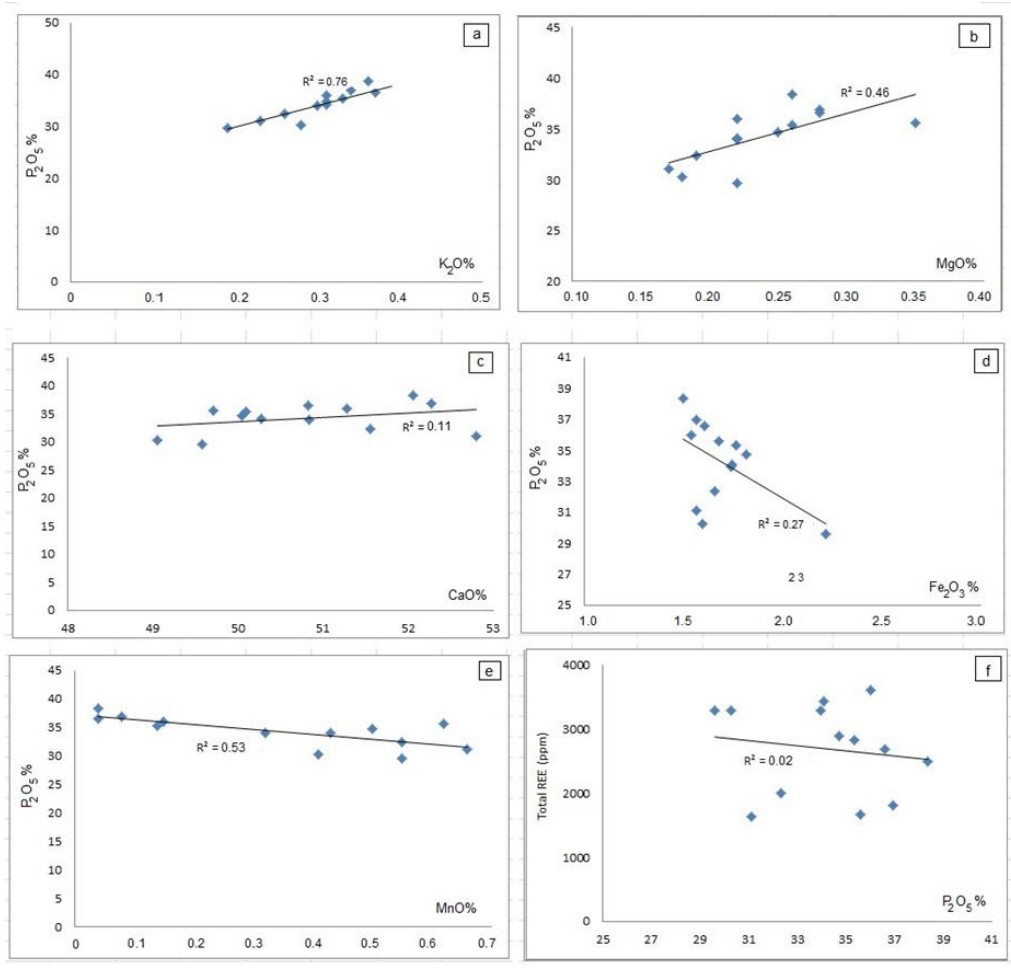

**Figure 5.** Binary major oxide plots: (**a**) P$_2$O$_5$ vs. K$_2$O; (**b**) P$_2$O$_5$ vs. MgO; (**c**) P$_2$O$_5$ vs. CaO; (**d**) P$_2$O$_5$ vs. Fe$_2$O$_3$; (**e**) P$_2$O$_5$ vs. MnO; (**f**) total REEs vs. P$_2$O$_5$.

### 4.2. Trace Element Geochemistry

Table 3 contains the results of the trace elements determined by ICP-MS at the Bureau Veritas Minerals (BVM) Laboratory. The mean trace elements contents for the phosphate nodule samples were 16 ppm for V, 32.7 ppm for Cr, 4.2 ppm for Co, 13.4 ppm for Ni, 75.8 ppm for Cu, 11.5 ppm for Rb, 395 ppm for Sr, 764 ppm for Y, 12.9 ppm for Zr, 0.7 ppm for Nb, 183 ppm for Ba, 0.09 ppm for Hf, 10.8 ppm for As, 0.26 ppm for Cd, 2.1 ppm for U, 1.7 ppm for Th, 84 ppm for Pb, 62.2 ppm for Zn, 0.13 ppm for Ta, and 3.1 ppm for Sc. In comparison to PAAS, the nodules were enriched in Cu, Cd, Sr, Y, As, Pb, Ni, V, Cr, Co, Zn, Hf, Zr, Nb, Ba, U, Th, and Rb [60] (Figure 6). Rb and Ba were lower compared to PAAS, whereas Sr, a large-ion lithophile element, was substantially enriched. Except for Y, all of the high-field-strength elements (HFSE; Nb, Hf, Th, Zr, U, and Y) were substantially lower than PAAS (Figure 6).

**Table 3.** Trace element concentrations (ppm) of the phosphate nodules.

| | V | Cr | Co | Ni | Cu | Rb | Sr | Y | Zr | Nb | Ba | Hf | As | Cd | U | Th | Pb | Zn | Th/U | V/Sc | V/Cr | Ni/Co |
|---|---|---|---|---|---|---|---|---|---|---|---|---|---|---|---|---|---|---|---|---|---|---|
| DL | 1 | 0.5 | 0.1 | 0.1 | 0.01 | 0.1 | 0.5 | 0.01 | 0.1 | 0.02 | 5 | 0.02 | 0.1 | 0.01 | 0.1 | 0.1 | 0.1 | 0.1 | | | | |
| PH-01 | 21 | 36 | 1.6 | 12.8 | 178 | 11.7 | 358 | 696 | 15.5 | 1.3 | 162 | 0.19 | 14.2 | 0.38 | 1.9 | 3.1 | 85 | 132 | 1.6 | 5 | 0.58 | 8.00 |
| PH-02 | 12 | 16 | 5.7 | 11.5 | 52 | 12.2 | 294 | 528 | 11.9 | 0.8 | 102 | 0.08 | 10.4 | 0.22 | 1.3 | 1.8 | 68 | 55 | 1.4 | 6 | 0.75 | 2.02 |
| PH-03 | 15 | 45 | 7.5 | 13.8 | 22 | 11.9 | 525 | 862 | 13.2 | 0.3 | 304 | 0.05 | 6.3 | 0.08 | 2.7 | 1.0 | 17 | 12 | 0.4 | 4 | 0.33 | 1.84 |
| PH-04 | 19 | 48 | 6.6 | 9.5 | 48 | 11.3 | 454 | 946 | 13.8 | 0.3 | 221 | 0.09 | 8.8 | 0.15 | 3.1 | 1.1 | 115 | 28 | 0.4 | 5 | 0.40 | 1.44 |
| PH-05 | 11 | 15 | 4.0 | 14.6 | 116 | 13.4 | 327 | 699 | 10.0 | 0.2 | 119 | 0.13 | 11.7 | 0.31 | 2.0 | 1.7 | 178 | 68 | 0.9 | 6 | 0.73 | 3.65 |
| PH-06 | 17 | 27 | 2.3 | 13.5 | 66 | 13.2 | 362 | 695 | 17.2 | 0.8 | 147 | 0.06 | 12.5 | 0.27 | 2.5 | 2.7 | 44 | 74 | 1.1 | 6 | 0.63 | 5.87 |
| PH-07 | 18 | 33 | 5.8 | 10.7 | 14 | 10.1 | 550 | 574 | 9.6 | 0.1 | 250 | 0.13 | 16.4 | 0.44 | 3.4 | 0.8 | 23 | 57 | 0.2 | 9 | 0.55 | 1.84 |
| PH-08 | 13 | 31 | 1.7 | 17.8 | 78 | 11.5 | 322 | 935 | 15.3 | 0.8 | 143 | 0.01 | 8.4 | 0.25 | 1.0 | 2.1 | 87 | 36 | 2.1 | 3 | 0.42 | 10.47 |
| PH-09 | 18 | 43 | 2.2 | 16.2 | 108 | 8.3 | 361 | 940 | 9.4 | 1.2 | 201 | 0.04 | 8.7 | 0.26 | 1.4 | 0.7 | 135 | 98 | 0.5 | 6 | 0.42 | 7.36 |
| PH-10 | 13 | 42 | 5.7 | 7.6 | 218 | 10.2 | 326 | 646 | 8.1 | 0.8 | 202 | 0.17 | 15.1 | 0.43 | 2.2 | 4.2 | 76 | 145 | 1.9 | 3 | 0.31 | 1.33 |
| PH-11 | 12 | 38 | 8.9 | 12.8 | 15 | 10.6 | 552 | 793 | 7.4 | 0.7 | 310 | 0.13 | 6.3 | 0.03 | 3.1 | 1.6 | 14 | 144 | 0.5 | 3 | 0.32 | 1.44 |
| PH-12 | 11 | 32 | 5.2 | 6.6 | 8 | 8.0 | 439 | 636 | 3.2 | 0.6 | 182 | 0.06 | 6.7 | 0.01 | 3.1 | 0.8 | 212 | 9 | 0.3 | 4 | 0.34 | 1.27 |
| PH-13 | 11 | 18 | 5.4 | 7.4 | 8 | 8.1 | 543 | 497 | 3.0 | 0.4 | 252 | 0.09 | 7.2 | 0.01 | 3.1 | 0.8 | 212 | 9 | 0.3 | 6 | 0.61 | 1.37 |
| Mean | 16 | 32.7 | 4.2 | 13.4 | 75.8 | 11.5 | 395 | 764 | 12.9 | 0.6 | 183 | 0.09 | 10.8 | 0.26 | 2.1 | 1.7 | 84 | 62 | 0.88 | 4.97 | 0.49 | 3.69 |

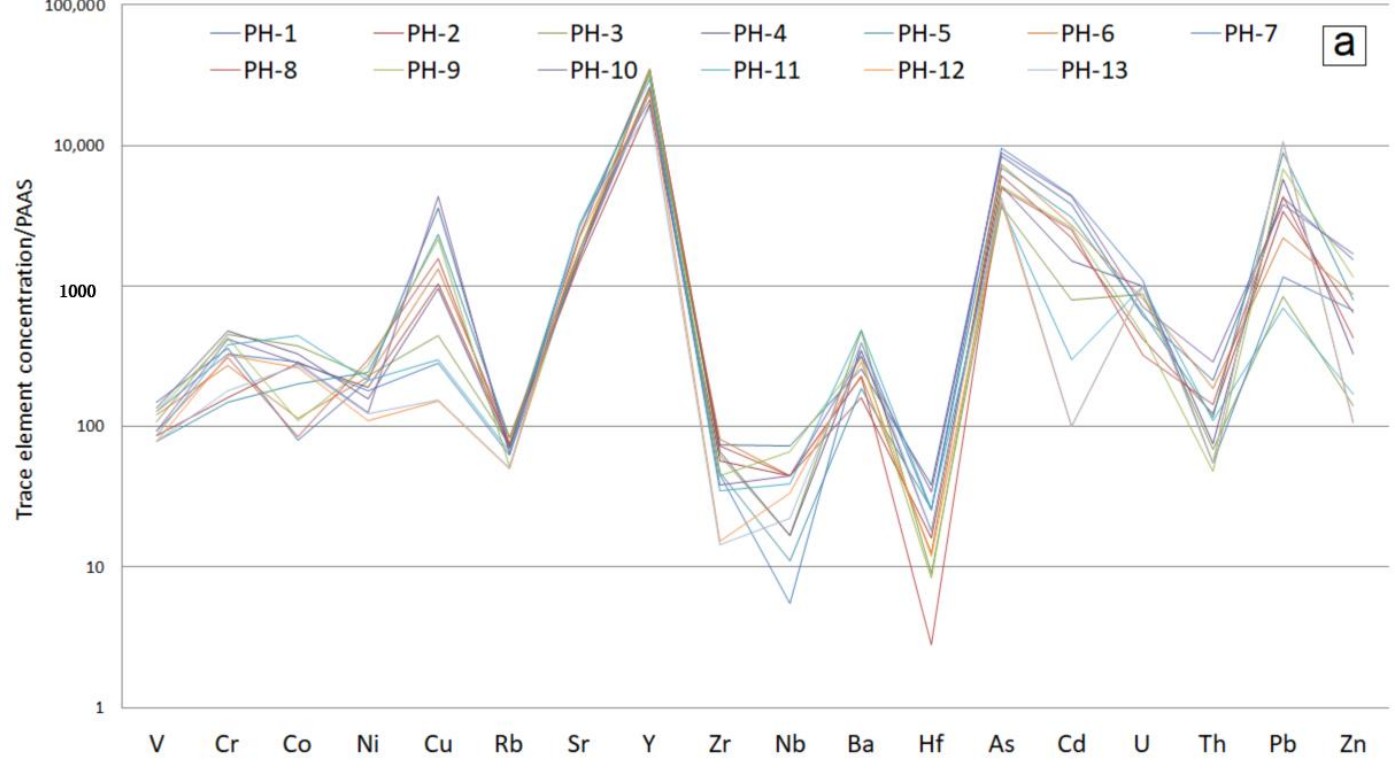

**Figure 6.** *Cont*.

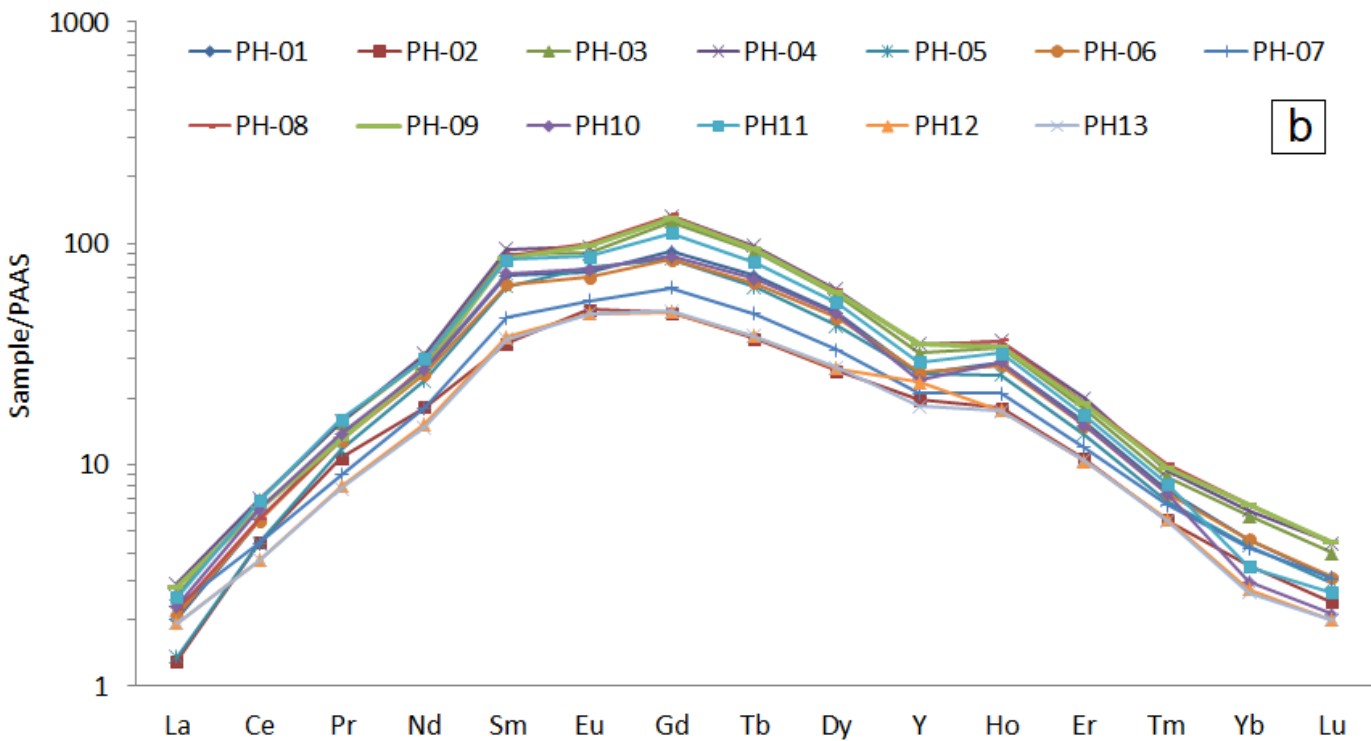

**Figure 6.** PAAS-normalized trace element (**a**) and REE (**b**) patterns of the phosphate nodules [60].

The Th/U ratios in the studied phosphate samples varied from 0.2 to 2.1. The Ni/Co ratios varied from 1.3 to 10.5. The V/Sc and V/Cr ratios were 4.97 (3 to 9) and 0.49 (0.31 to 0.75), respectively (Tables 3 and 4). The oxic/suboxic relationship between redox conditions and the precipitation of phosphate nodules is shown by these element ratios. The oxidation level of the water column affected some elements' solubility. According to the redox status of the water column, these elements may therefore be enriched or deficient in marine sediments. Low dissolved oxygen concentrations and moderate H2S concentrations are characteristics of suboxic settings. The absence of dissolved oxygen and the presence of $H_2S$ are characteristics of anoxic settings [64]. The redox environment of the paleo-ocean can be determined by redox-sensitive replacements (Ni/Co, Th/U, V/Cr, V/Sc, and REEs) [65]. In addition, the enrichment of both Mo ($>5$ µg g$^{-1}$) and V ($>23$ µg g$^{-1}$), with V not exceeding 46 µg g$^{-1}$, provides compelling evidence for a euxinic basin-type depositional environment [66], while the enrichment of V ($>46$ µg g$^{-1}$), U ($>5$ µg g$^{-1}$), and Mo ($>5$ µg g$^{-1}$) serves as robust evidence for sediment deposition [67]. Conversely, U enrichment ($>1$ µg g$^{-1}$/%), consistent with the low enrichment of V ($<23$ µg g$^{-1}$) and Mo ($<5$ µg g$^{-1}$), provides strong evidence of sediment precipitation in oxic waters [13,68]. Furthermore, both U and V were readily incorporated into the crystal structure of apatite, and this process continued during diagenesis [69], Suboxic conditions are characterized by low dissolved oxygen content (0.2–0.0 mL L$^{-1}$), while anoxic conditions are characterized by the absence of oxygen [70,71]. The presence of free sulfides in the water column indicates the precipitation of anoxic sediments and the presence of sulfidic/euxinic conditions. In addition, dissolved sulfites are not found in suboxic environments.

**Table 4.** Redox classification of the depositional environment.

| Indicator | Oxic | Suboxic | Anoxic | Euxinic | Studied Nodules |
|---|---|---|---|---|---|
| $H_2S$ | | No free $H_2S$ in the water column | | Free $H_2S$ present in the water column | |
| $O_2$ concentration in bottom waters [a] ($mLO_2/LH_2O$) | $O_2 > 2$ | $0.2 < O_2 < 2$ | $O_2 < 0.2$ | $O_2 = 0$ | |
| Th/U [b] | >7 | 2–7 | 0–2 | - | 0.88 |
| V/Cr [c] | <2 | 2–4.25 | >4.25 | >4.25 | 0.49 |
| Ni/Co [c] | <5 | 5–7 | >7 | - | 3.69 |
| V/Sc [d] | - | - | - | >24 | 4.97 |

[a] [71], [b] [64], [c] [72], [d] [73].

### 4.3. Rare-Earth Element Geochemistry

Table 5 lists the concentrations of rare-earth elements present in the phosphate nodules. The REE concentrations of nodule samples varied from 1638 ppm to 3602 ppm (mean: 2684 ± 62 ppm), except for Y. Figure 6 depicts both PAAS-normalized trace element and REE patterns of the phosphate nodules according to Taylor and McLennan [60]. When the REE values of the nodules were normalized with PAAS, the medium REEs were enriched in the nodules, while the light and heavy REEs were depleted. The normalized average REE abundances are as follows: medium REEs (350) were followed by heavy (58) then light REEs (44.8) (Figure 6). Consistent with the REE trend observed in the study area (Figure 7), previous studies by Emsbo et al. [74] and Yu et al. [75] have proposed that the formation of MREE-rich phosphates is influenced by variations in seawater composition across different time periods and regions (Figure 7). The $La_n/Yb_n$ ratios showed the enrichment patterns between heavy and light REEs. The $La_n/Yb_n$ values of the phosphate nodules ranged between 0.31 and 0.78 with a mean of 0.52 ± 0.02 (Table 5), which is comparable to the corresponding values of modern seawater (0.2–0.5; [76]) in the phosphate nodules, verifying the HREE enrichments [77]. Low $La_n/Yb_n$ ratios in phosphate nodules indicate that REE concentration is associated with both the adsorption of REEs during the evolution of these nodules and the substitution mechanism by recrystallization [77]. The $La_n/Ho_n$ ratios changed from 0.05 to 0.11 with a mean of 0.08 ± 0.01 (Table 5), and this also indicates that the MREE contents of the phosphate nodules were enriched more than the LREE contents. The $(Sm/Pr)_n$ and $(Sm/Yb)_n$ values show that all of the studied phosphate nodules had MREE enrichment (Figure 8). Looking at Figure 8, the phosphate nodules of Podillya, Sonrai, and Gorgan were enriched by MREEs, while the Yangtze deposits were depleted by MREEs. Tebessa, Alborz, and Hazm Al-Jalamid deposits were enriched in HREEs. One significant finding of this study is that the patterns of rare-earth elements are specific to certain time periods, which indicates the geological age of the deposits. This discovery serves as a robust predictive tool for exploring high-REE deposits [74].

**Table 5.** Rare-earth element concentrations (ppm) of phosphate nodules.

| | Y | La | Ce | Pr | Nd | Sm | Eu | Gd | Tb | Dy | Ho | Er | Tm | Yb | Lu | REE | (Ce/Ce*)n | (Eu/Eu*)n | (Y/Y*)n | (Pr/Pr)n | (Sm/Yb)n |
|---|---|---|---|---|---|---|---|---|---|---|---|---|---|---|---|---|---|---|---|---|---|
| DL | 3 | 0.5 | 0.1 | 0.02 | 0.02 | 0.02 | 0.02 | 0.02 | 0.02 | 0.02 | 0.02 | 0.02 | 0.02 | 0.02 | 0.02 | | | | | | |
| PH-01 | 696 | 77 | 448 | 118 | 907 | 396 | 80 | 428 | 55 | 227 | 29 | 44 | 3.2 | 13 | 1.33 | 2825 | 0.74 | 0.91 | 0.67 | 0.82 | 16 |
| PH-02 | 528 | 50 | 354 | 94 | 618 | 195 | 54 | 226 | 28 | 124 | 18 | 30 | 2.3 | 10 | 1.03 | 1805 | 0.74 | 1.20 | 0.88 | 0.94 | 10 |
| PH-03 | 862 | 100 | 547 | 137 | 1027 | 494 | 97 | 581 | 70 | 273 | 33 | 50 | 3.6 | 17 | 1.72 | 3431 | 0.76 | 0.84 | 0.69 | 0.84 | 15 |
| PH-04 | 946 | 111 | 558 | 138 | 1074 | 520 | 104 | 616 | 75 | 291 | 36 | 56 | 3.9 | 17 | 1.88 | 3602 | 0.76 | 0.85 | 0.71 | 0.81 | 15 |
| PH-05 | 699 | 52 | 348 | 103 | 828 | 356 | 84 | 398 | 49 | 198 | 25 | 39 | 2.8 | 12 | 1.26 | 2497 | 0.68 | 1.05 | 0.76 | 0.82 | 15 |
| PH-06 | 695 | 80 | 446 | 114 | 868 | 360 | 76 | 393 | 51 | 213 | 28 | 43 | 3.0 | 13 | 1.33 | 2689 | 0.75 | 0.94 | 0.70 | 0.83 | 14 |
| PH-07 | 574 | 87 | 349 | 82 | 616 | 253 | 59 | 295 | 37 | 154 | 20 | 34 | 2.7 | 12 | 1.31 | 2000 | 0.78 | 1.01 | 0.78 | 0.80 | 11 |
| PH-08 | 935 | 83 | 459 | 117 | 933 | 490 | 107 | 619 | 73 | 284 | 35 | 54 | 4.0 | 19 | 1.90 | 3279 | 0.75 | 0.90 | 0.72 | 0.77 | 13 |
| PH-09 | 940 | 106 | 505 | 116 | 922 | 470 | 106 | 606 | 72 | 279 | 34 | 53 | 4.0 | 19 | 1.94 | 3294 | 0.81 | 0.91 | 0.74 | 0.78 | 13 |
| PH-10 | 646 | 86 | 502 | 122 | 926 | 404 | 83 | 405 | 53 | 225 | 29 | 44 | 3.0 | 8 | 0.91 | 2891 | 0.78 | 0.95 | 0.62 | 0.83 | 25 |
| PH-11 | 793 | 96 | 553 | 138 | 1010 | 467 | 94 | 516 | 63 | 251 | 32 | 47 | 3.4 | 10 | 1.14 | 3282 | 0.75 | 0.89 | 0.67 | 0.87 | 24 |
| PH-12 | 636 | 74 | 294 | 70 | 516 | 210 | 52 | 230 | 29 | 128 | 17 | 30 | 2.3 | 8 | 0.86 | 1662 | 0.76 | 1.11 | 1.05 | 0.85 | 14 |
| PH-13 | 497 | 73 | 294 | 69 | 501 | 205 | 52 | 229 | 29 | 128 | 17 | 30 | 2.3 | 7 | 0.86 | 1638 | 0.76 | 1.10 | 0.82 | 0.84 | 14 |
| Avrg | 727 | 82 | 435 | 109 | 827 | 371 | 81 | 425 | 53 | 214 | 27 | 43 | 3.1 | 13 | 1.34 | 2684 | 0.76 | 0.94 | 0.74 | 0.83 | 15 |

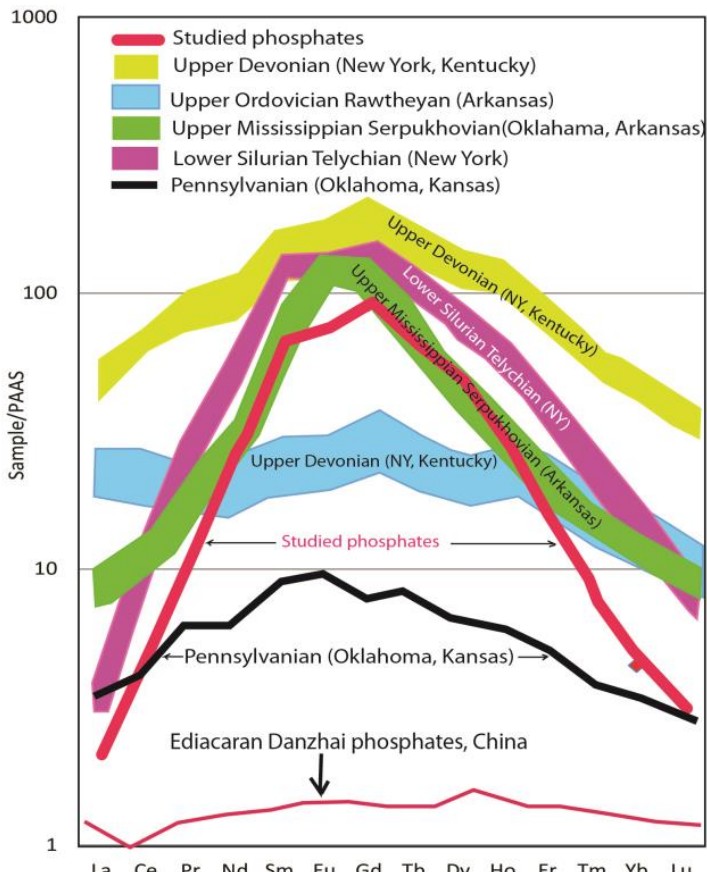

**Figure 7.** Comparison of PAAS-normalized REE patterns in different sedimentary phosphates. Data from [74,75].

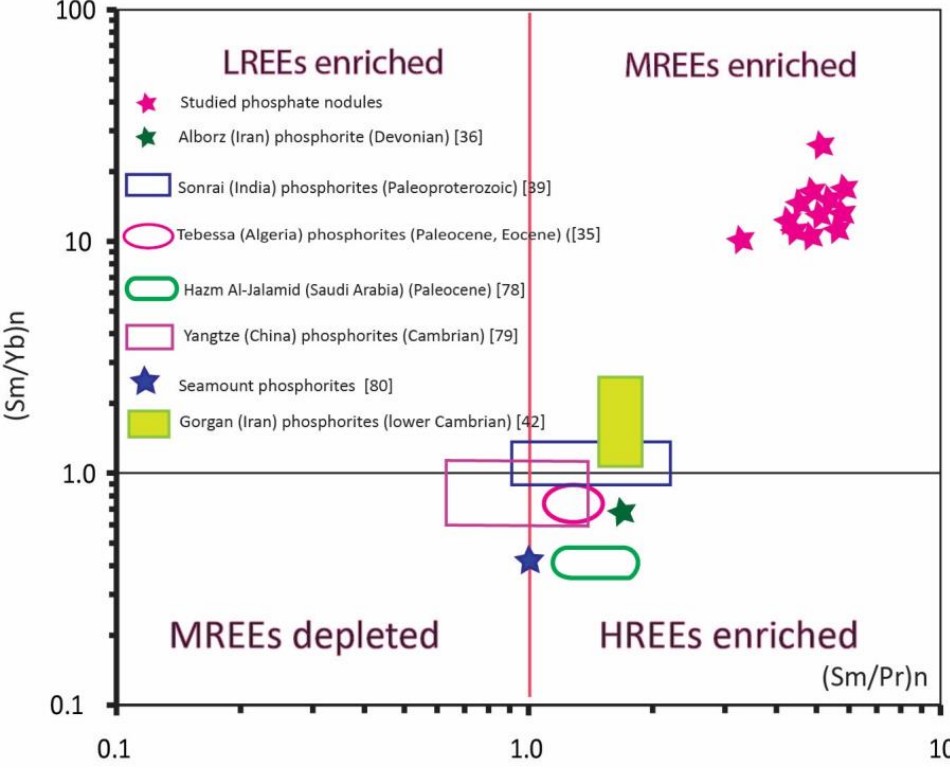

**Figure 8.** $(Sm/Pr)_n$ and $(Sm/Yb)_n$ relationships in the phosphate nodules (taken from [36,78–80]).

In order to study the redox potentials of various sedimentary environments, cerium is the most helpful rare-earth element. Ce actually oxidizes quickly and is continuously and irrevocably eliminated from seawater's oxide minerals [81]. Therefore, a positive Ce anomaly in sedimentary basins indicates hydrogenetic deposition. Ce mostly enriches in hydrogenetic deposits due to irreversible Ce accumulation from seawater, but this is lower in hydrothermal and diagenetic nodules [82]. The precipitation conditions of the depositional environment, such as the temperature, pH, and $fO_2$, are found using the Eu/Eu* and Ce/Ce* ratios [83–85]. Under seawater pH and Eh conditions, Ce is primarily $Ce^{4+}$ and has very poor solubility. As a result of the seawater's severe depletion of Ce, it precipitated as $Ce^{4+}$ ($CeO_4$). Ce can exist in both +3 and +4 oxidation states. Only oxic environments experience this [86,87]. A negative Ce anomaly indicates well-oxygenated modern oceans, and this also points out the rapid deposition of oxide minerals [88]. The Ce/Ce* values of the examined phosphate nodules ranged from 0.68 to 0.81 and had significant negative Ce anomalies (Table 5).

The positive Eu anomalies indicate low oxygen fugacity in the hydrothermal waters and the existence of $Eu^{2+}$ during deposition. The phosphate was precipitated from solutions with increasing $fO_2$ or coprecipitated together with Eu-enriched phases or decreasing temperature. Eu anomalies can be associated with factors such as a decrease in temperature or increases in in $fO_2$ and pH [89]. Whereas positive Eu-anomalies are indicative of (extremely) reductive conditions, negative Eu-anomalies do not form under oxic conditions. Under normal surface conditions (i.e., seawater or pore water), the Eu/Eu* ratios do not change significantly, and the negative Eu-anomaly is likely due to other factors such as volcanic ash intrusion [35,90–93]. The average $(Eu/Eu^*)_n$ anomaly of the phosphate nodules in the study area was close to 1 (mean: $0.94 \pm 0.03$) (Table 5). Five of the phosphate nodules had a slightly positive Eu anomaly (>1); eight of the phosphate nodules were lower than 1 and had a slightly negative Eu anomaly.

In contrast to anoxic conditions, the oxic water samples show a REY trend with heavier REE enrichment and a negative Y anomaly. In comparison to the oxic saltwater above, the anoxic hypersaline brine waters in the Tyro sub-basin exhibited a negative Y anomaly [94]. The Y contents in the phosphate nodules changed from 497 to 946 ppm, with an average of $727 \pm 43$ ppm. The phosphate nodules had negative Y anomalies, except for one sample between 0.62 and 1.05 with a mean of $0.74 \pm 0.02$ (Table 4). These data also indicate that the phosphate nodules not only formed in environments where anoxia prevailed but also where episodes of oxic conditions could occur. Similarly, Chen et al. [95] reported that the ferromanganese oxides from 4071m water depth in the Gagua Ridge had negative Y anomalies. Lumiste et al. [96] indicated that the negative Y-anomalies were inherited from Fe-Mn oxides that delivered the REE to the pore water.

### 4.4. Source of REEs in Phosphate Nodules

According to Emsbo et al. [74], REE levels in phosphate nodules are typically homogeneous over the course of a geologic time, although they can vary between geologic times. Rare-earth elements are only a little fractionated, as evidenced by the low REE abundance and patterns of contemporary phosphate rocks that resemble those of contemporary saltwater [97]. The Ediacaran phosphates studied in this work were observed to have more REE contents than those in different geologic ages, except for Upper Silurian, Upper Devonian, and Upper Mississippian phosphates [36]. Several scientists believe that the primary REE content of phosphates is closely related to that of current saltwater because geologic processes have such strong control over REE content [45,98]. REE abundance variations have been linked to facies, particle size, depth, and the duration of precipitation [99,100]. REE concentrations in phosphate nodules have been formed as a direct product of ocean chemistry [45,76,101]. As an alternative, it is thought that the extremely high REE concentrations in phosphates are a byproduct of the REEs being redistributed between the authigenic and detrital phases of the phosphate during its diagenetic equilibration with pore water [100–102].

*4.5. Origin and Genetic Model of the Phosphate Nodules*

At the continental edges during various periods, marine phosphates produced irregular masses, nodules, sands, oolites, and pellets [103]. Burnett [103] found that pore fluids contained higher phosphate levels than bottom waters, contrary to some scientists' theories that phosphate production was directly caused by the inorganic precipitation of phosphates from seawater. Hence, it has been proposed that phosphate precipitation in the pore fluids of anoxic sediments may be the source of the development of marine phosphates. For phosphate to develop, dissolved phosphates must be present in the pore fluids. According to Stumm and Leckie [104], there are different mechanisms to supply phosphate ions to anoxic pore waters: (1) the decomposition of phosphorus-containing organic materials, (2) the reduction of hydrous ferric oxides that bind phosphate to their surfaces under reducing conditions, and (3) a possible pathway for modifying pore water phosphate concentrations is polyphosphate hydrolysis by Large Sulfur Bacteria. Relative to shale-normalized values and the bulk REE content of contemporary saltwater, the REE concentration in phosphates is up to 50–100 times greater [105]. Direct precipitation of phosphates from saltwater is therefore implausible. By far, the most significant source of REE in phosphate nodules, according to Felitsyn and Morad [105], is organic matter. They also agreed that the direct ejection of REEs from marine pore fluids is the most significant mechanism to account for the REE enrichment in authigenic phosphate. The possibility of hydrothermal fluids being sources of phosphate and REE has also been raised [106]. In addition to hydrogenous and hydrothermal sources, pore water diagenesis is an important source for REE in phosphates [107].

In Figure 9, a genetic model is suggested to account for the development of phosphate nodules. The model shown in this figure suggests that seawater can be divided into different layers based on redox conditions, including the top oxic layer (oxygen content: 2.0–8.0 mL/L), the dysoxic layer (oxygen content: 0.2–2.0 mL/L), the suboxic layer (oxygen content: 0.0–0.2 mL/L), and the bottom anoxic layer (oxygen content: 0.0 mL/L) [71]. In a deeper, organic-rich basin with anoxic/suboxic conditions, the phosphate nodules precipitated on slopes in relation to the strong upwelling stage [108]. The majority of the organic material that has accumulated at the seafloor's sediment–water interface was first oxidized before phosphate was transported there with the sediment pore water [109]. Iron/manganese oxides simultaneously adsorbed phosphorus as it flowed up slopes with a benthic flux that contained considerable levels of trace elements and REEs [110]. The phosphate nodules formed by the combination of the Fe-Mn oxides in the benthic flow with the REEs and P ultimately contained a significant amount of REE phases, trace elements, and $PO_4^{3-}$ [10].

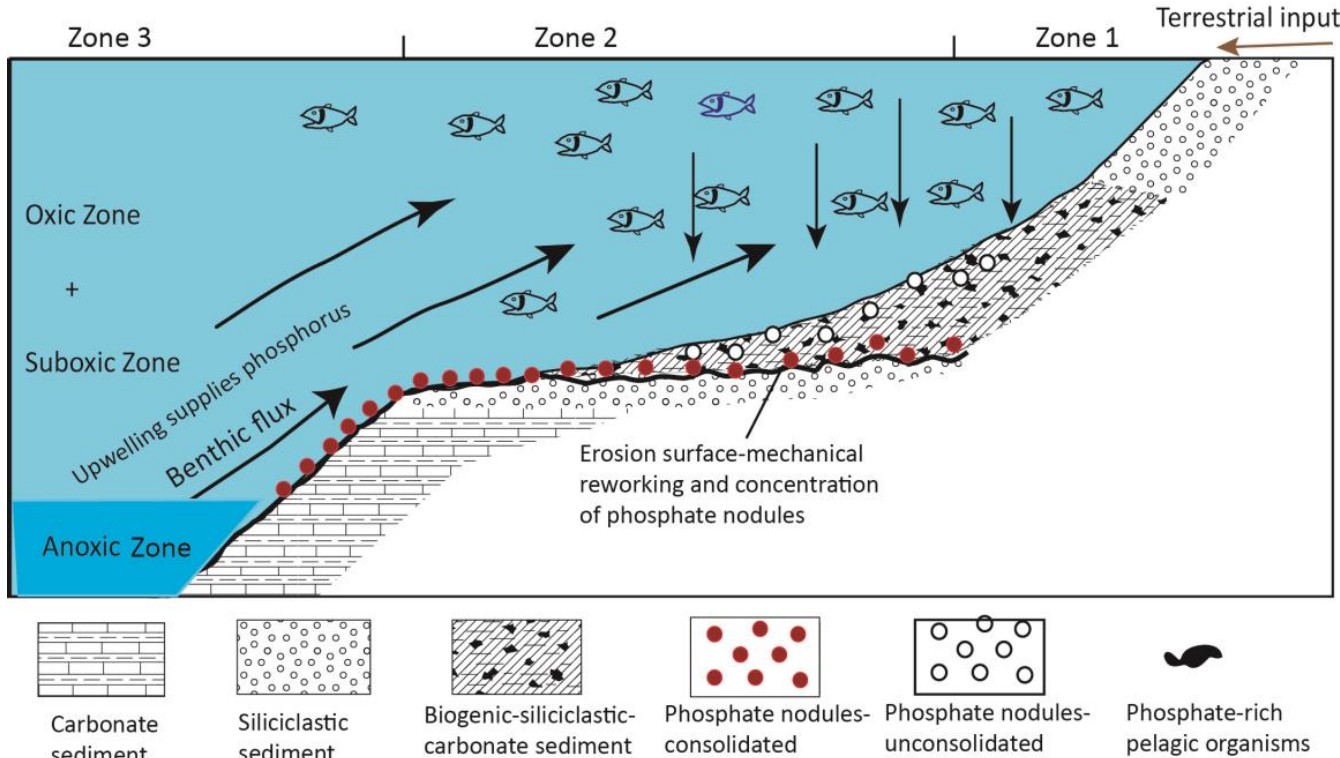

**Figure 9.** Genetic model for the phosphate nodules formed in areas of upwelling on open ocean shelves. Zone 1 forms from near-shore, shallow-water siliciclastic deposits. Zone 2 contains both phosphate-rich biogenic detritus in sediments and phosphate nodules formed in this zone by diagenetic processes. Zone 3 consists of carbonate sediments containing local phosphate nodules in the deeper water zone. Data from [111,112].

## 5. Conclusions

The Volyn-Podillya-Moldavia Basin, located in the southwest of Ukraine at the edge of the Ukrainian crystalline shield, is part of an extensive system of Neoproterozoic-Paleozoic sedimentary basins that developed along the western slope of the East European Craton. It is covered by younger Paleozoic rocks; their outcrops extend to the western part of the basin. The major oxide contents mainly consist of CaO and $P_2O_5$, and lower $SiO_2$, $Fe_2O_3$, $Al_2O_3$, MnO, $K_2O$, MgO, $Na_2O$, $TiO_2$, and $Fe_2O_3$. The total trace element contents in the studied phosphate nodules varied from 1638 and 3602 ppm. The phosphate nodules contained more Sr, Cu, Y, Cd, As, and Pb and lower Cr, V, Co, Ni, Nb, Zr, Rb, Ba, Hf, U, Zn, and Th compared to PAAS. Some trace element ratios mainly indicate the oxic and suboxic zones for the precipitation of the phosphate nodules. The nodules had very high total REE concentrations, between 1638 and 3602 ppm, and have been enriched mostly in regard to medium REEs, and less to heavy REEs, compared to PAAS. All nodules showed negative Ce, Y, and Pr anomalies and mostly negative Eu anomalies. These anomalies further indicate oxic and suboxic conditions during nodule formation. We propose the following genetic model for phosphate nodule formation and indicate that our phosphorite was formed by early diagenetic reactions starting at the sediment–water interface. Seawater was separated into three layers based on redox conditions: the bottom anoxic zone, the suboxic zone, and the oxic zone. The vigorous upwelling from an organic-rich region in deeper seas under anoxic/suboxic conditions caused the phosphate nodules to form on slopes. Upwelling is a great source of nutrients. This permitted an increase in bioproductivity and then the accumulation of organic matter. The biodegradation of this organic matter permitted the P-release, which was incorporated inside the nodules during the diagenetic process. The majority of the organic material that settled at the seafloor's sediment–water interface was first oxidized, and then phosphate filled the sediment's pore water by migrating to the

water in the upper zone. The most important source of REE in phosphate nodules was organic matter, and this REE was enriched in authigenic phosphate as a result of the direct evacuation of REE from marine pore fluids. In addition to hydrogen and hydrothermal sources, pore water diagenesis is among the important sources of REE in phosphates.

**Author Contributions:** Conceptualization, A.S.; methodology, A.S., Y.S., E.Z., N.K. and A.E.A.; investigation, A.S. and B.S.; resources, Y.S. and A.E.A.; writing—original draft preparation, A.S. and A.E.A.; writing—review and editing, A.S. and A.E.A.; visualization, A.S. and A.E.A. All authors have read and agreed to the published version of the manuscript.

**Funding:** This study was funded by Firat University with project number MF.22.28.

**Data Availability Statement:** The data are available from the authors upon reasonable request.

**Conflicts of Interest:** The authors declare that they have no known competing financial interests or personal relationships that could have appeared to influence the work reported in this paper.

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
