# Peer review of "Geochemical Evidence of Ediacaran Phosphate Nodules in the Volyno-Podillya-Moldavia Basin, Ukraine"

_minerals, doi:10.3390/min13040539_

Round 1

Reviewer 1 Report

The authors analyzed the primary oxide, trace, and rare earth element compositions of the phosphate nodules in the Volyno-Podillya-Moldavia Basin, Ukraine, and discussed the geochemical feature of phosphate nodules and thought the formed condition of nodules is under oxic and suboxic sea conditions. At last , the Authors created the genetic model of phosphate nodules,and thought nodules were precipitated on slopes in response to a significant 34 upwelling from a deeper basin with abundant organic matter under anoxic/suboxic conditions. The research has significance for the genetic of phosphate nodules.

However, there are still some deficiencies in the article:

line 237 and line 238, the total concentration of trace elements is insignificance

In addition, supply the material in paper as following:

 (1) In figure 3. add the explanations.(see the attachment)

a. b. ,C.

(2) In analytical method, add the experiment condition, for example, temperation et al.,

Some mistakes in the paper as following:

(1) Line 4 and 5: Auhtors "Ahmet Sasmaz1*, Bilge Sasmaz1 and Yevheniia Soldatenko2 Abder El Albani3, Edward Zhovinsky3, Nataliya Kryuchenko4 "modifed "Ahmet Sasmaz1*, Bilge Sasmaz1 , Yevheniia Soldatenko2 ,Abder El Albani3, Edward Zhovinsky3, Nataliya Kryuchenko4"

(2) Citation:  the authors add  one authors, "Soldatenko,Y." after “Sasmaz, B.; ”  

3line 89: "Yaoming et al. 2004" revised "Yang et al. 2004

(4) line 277:"Taylor and McLennan, 1995" revised “Taylor and McLennan, 1985”

5line 306: “Takahashi et al., 2007”undiscovered in referrence;

6Line 326:" Deng et al.2014." undiscovered in referrence;

(7) among referrences:

a.line 491,line522 , line 524,line 598: ref.22,ref.37,ref.38,and ref.71, year among bracket modify as year,for example,(2012)modify as 2012.

b.line 505: ref. 29 Author Emsho is error. It is Emsbo.

c.line 576,line 579,line 583,line 586: ref.62 and ref.63 and ref.65 and ref.66 arrange again.

d.line 666:ref.102, Yang,Y., Tu,G., Hu,R., 2004. REE and trace element geochemistry of Yinachang Fe-Ca-REE deposit, Yunuan Province. China. Chinese J Geochem. 23, 265–274

Author Response

Responses to Reviewer 1

The authors analyzed the primary oxide, trace, and rare earth element compositions of the phosphate nodules in the Volyno-Podillya-Moldavia Basin, Ukraine, and discussed the geochemical feature of phosphate nodules, and thought the formed condition of nodules is under oxic and suboxic sea conditions. At last , the Authors created the genetic model of phosphate nodules,and thought nodules were precipitated on slopes in response to a significant 34 upwelling from a deeper basin with abundant organic matter under anoxic/suboxic conditions. The research has significance for the genetic of phosphate nodules.

However, there are still some deficiencies in the article:

line 237 and line 238, the total concentration of trace elements is insignificance

 it was corrected

In addition, supply the material in paper as following:

 (1) In figure 3. add the explanations.(see the attachment)

It was added the explanations of Sp A on Fig. 3 as Continuous/discontinuous fossil levels

(2) In analytical method, add the experiment condition, for example, temperation et al.,

 This study is not an experimental study and there is no temperature

Some mistakes in the paper as following:

(1) Line 4 and 5: Auhtors "Ahmet Sasmaz1*, Bilge Sasmaz1 and Yevheniia SoldatenkoAbder El Albani3, Edward Zhovinsky3, Nataliya Kryuchenko4 "modifed "Ahmet Sasmaz1*, Bilge Sasmaz1, Yevheniia Soldatenko,Abder El Albani3, Edward Zhovinsky3, Nataliya Kryuchenko4"

It was corrected

(2) Citation:  the authors add  one authors, "Soldatenko,Y.;" after “Sasmaz, B.; ”  

(3)line 89: "Yaoming et al. 2004" revised "Yang et al. 2004

It was corrected

(4) line 277:"Taylor and McLennan, 1995" revised “Taylor and McLennan, 1985”

It was corrected

(5)line 306: “Takahashi et al., 2007”undiscovered in reference;

It was added to references

(6)Line 326:" Deng et al.2014." undiscovered in referrence;

It was added to references

(7) among referrences:

All references were corrected to Minerals’s format as (2012)

a.line 491,line522 , line 524,line 598: ref.22,ref.37,ref.38,and ref.71, year among bracket modify as year,for example,(2012)modify as 2012.

b.line 505: ref. 29 Author Emsho is error. It is Emsbo.

It was corrected as Emsbo

c.line 576,line 579,line 583,line 586: ref.62 and ref.63 and ref.65 and ref.66 arrange again.

d.line 666:ref.102, Yang,Y., Tu,G., Hu,R., 2004. REE and trace element geochemistry of Yinachang Fe-Ca-REE deposit, Yunuan Province. China. Chinese J Geochem. 23, 265–274

It was added to references

Reviewer 2 Report

Overview and general recommendation:

This study examines the Ediacaran phosphate nodules of the Volyno-Podillya-Moldavia Basin, primarily focusing on the REE and trace element systematic within the phosphate nodules. The overall quality of this manuscript is poor. The readability and comprehensibility of this manuscript need serious improvement. I have significant issues with the theoretical background and the interpretations provided by the author of the manuscripts. Throughout the manuscript, the authors make erroneous statements, some of which are listed below. One of my main issue is using bimetal ratios and Ce-anomalies to constrain paleo-redox conditions. These methods should not be used in sedimentary phosphorites. Furthermore, the interpretations regarding the formational environment are based on negative Ce-anomalies. However, I am uncertain if there are true negative Ce-anomalies in these nodules (sensu Bau and Dulski, 1996).

Some of my comments and recommendations are listed in detail below. In its current form, I recommend rejection, based on the poor readability of the manuscript, poor theoretical background and shaky interpretations provided by the authors. 

Comments:

Lines 42-43 I suggest rewording this sentence for clarity. What are the different stages?

Lines 81-82 This statement is confusing. Whereas phosphorites contain significant amounts of REE, a close relationship between “phosphate deposits” and REE “during precipitations” is not strictly true. Nascent apatite contains very low concentrations of REE and the vast majority of these elements are taken up during diagenesis.

Line 111 overly -> overlie (?)

Lines 170-172 Needs rewording

Line 237-238 Total trace element concentrations? Total measured trace element concentrations would be more accurate but I am unsure what these concentration values are meant to indicate.

Lines 257-269 The use of bimetal ratios  with universal threshold values (i.e., V/Cr and Th/U) has been shown to be an unreliable indicator of past seawater conditions (Algeo and Liu, 2020). I would advise that the authors use enrichment factors (Algeo and Liu, 2020) or trace-metal/aluminium normalization (Bennett and Canfield, 2020) instead. Furthermore, both U and V are readily incorporated into the crystal structure of apatite (Pan and Fleet, 2002) and this process continues during diagenesis. Using U- and V-based proxies in sedimentary phosphorites can lead to erroneous conclusions regarding the redox state of the formational environment (Lumiste et al., 2021).  

Lines 264-267 Suboxic conditions are defined by low amounts of dissolved oxygen (0.2-0.0 ml/1), and anoxic conditions are defined by the absence of oxygen (e.g., Tribovillard et al., 2006; Tyson and Pearson, 1991). Anoxic sediments can be further divided into sulfidic/euxinic conditions if there are free sulfides in the water column. Suboxic environments do not contain dissolved sulfides.

Table 5 Are these numbers concentrations? Perhaps these are PAAS-normalize values and not concentrations (ppm)?

Line 318-320 These are not true negative Ce anomalies, the negative Ce/Ce* values noted by the authors are likely caused by high Pr concentrations. I advise that the authors used the method proposed by Bau and Dulski, (1996) for classifying Ce-anomalies.

Lines 318-344 I have serious issues with this discussion. Using REE concentrations in sedimentary phosphorites to reconstruct past seawater conditions is problematic. During precipitation, sedimentary apatite contains very low concentrations of REEs, and the vast majority of these elements are taken up from pore water during diagenesis.

Line 366-376 “It is suggested two mechanisms to provide phosphate ions from anoxic pore waters by Stumm and Leckie (1970).” What are the mechanisms? The formation of phosphorites is not described.

References:

Algeo, T.J., Liu, J., 2020. A re-assessment of elemental proxies for paleoredox analysis. Chem. Geol. 119549. https://doi.org/10.1016/j.chemgeo.2020.119549

Bau, M., Dulski, P., 1996. Distribution of yttrium and rare-earth elements in the Penge and Kuruman iron-formations, Transvaal Supergroup, South Africa. Precambrian Res. 79, 37–55. https://doi.org/10.1016/0301-9268(95)00087-9

Bennett, W.W., Canfield, D.E., 2020. Redox-sensitive trace metals as paleoredox proxies: A review and analysis of data from modern sediments. Earth-Science Rev. 204, 103175. https://doi.org/10.1016/j.earscirev.2020.103175

Lumiste, K., Mänd, K., Bailey, J., Stüeken, E.E., Paiste, K., Lang, L., Sepp, H., Lepland, A., Kirsimäe, K., 2021. Constraining the conditions of phosphogenesis: Stable isotope and trace element systematics of Recent Namibian phosphatic sediments. Geochim. Cosmochim. Acta 302, 141–159. https://doi.org/10.1016/j.gca.2021.03.022

Pan, Y., Fleet, M.E., 2002. Compositions of the Apatite-Group Minerals: Substitution Mechanisms and Controlling Factors. Rev. Mineral. Geochemistry 48, 13–49. https://doi.org/10.2138/rmg.2002.48.2

Tribovillard, N., Algeo, T.J., Lyons, T., Riboulleau, A., 2006. Trace metals as paleoredox and paleoproductivity proxies: An update. Chem. Geol. 232, 12–32. https://doi.org/10.1016/j.chemgeo.2006.02.012

Tyson, R. V., Pearson, T.H., 1991. Modern and ancient continental shelf anoxia. Mod. Anc. Cont. shelf anoxia 1–24. https://doi.org/10.2307/3515153

Author Response

Responses to Reviewer 2

Overview and general recommendation:

This study examines the Ediacaran phosphate nodules of the Volyno-Podillya-Moldavia Basin, primarily focusing on the REE and trace element systematic within the phosphate nodules. The overall quality of this manuscript is poor. The readability and comprehensibility of this manuscript need serious improvement. I have significant issues with the theoretical background and the interpretations provided by the author of the manuscripts. Throughout the manuscript, the authors make erroneous statements, some of which are listed below. One of my main issue is using bimetal ratios and Ce-anomalies to constrain paleo-redox conditions. These methods should not be used in sedimentary phosphorites. Furthermore, the interpretations regarding the formational environment are based on negative Ce-anomalies. However, I am uncertain if there are true negative Ce-anomalies in these nodules (sensu Bau and Dulski, 1996).

Some of my comments and recommendations are listed in detail below. In its current form, I recommend rejection, based on the poor readability of the manuscript, poor theoretical background and shaky interpretations provided by the authors. 

Comments:

Lines 42-43 I suggest rewording this sentence for clarity. What are the different stages?

It was added this sentence

Over the last 3.85 billion years of Earth's history, various geological events have occurred in distinct periods. Notably, atmospheric oxygen levels rose to values higher than 0.2 atm during the Proterozoic Eon, specifically between 0.80 and 0.54 billion years ago. This increase in oxygen was accompanied by a similar trend in the shallow oceans

Lines 81-82 This statement is confusing. Whereas phosphorites contain significant amounts of REE, a close relationship between “phosphate deposits” and REE “during precipitations” is not strictly true. Nascent apatite contains very low concentrations of REE and the vast majority of these elements are taken up during diagenesis.

 This sentence was removed from Introduction.

Line 111 overly -> overlie (?)

It was corrected

Lines 170-172 Needs rewording

It was corrected ‘’The samples, which varied in size, were divided into two sections: one was used to element analysis, while the other was utilized for mineralogical and petrographic investigations.’’

Line 237-238 Total trace element concentrations? Total measured trace element concentrations would be more accurate but I am unsure what these concentration values are meant to indicate.

Table 3 contains the results of the trace elements determined by ICP-MS at at the Bureau Veritas Minerals (BVM) Laboratories.

Lines 257-269 The use of bimetal ratios  with universal threshold values (i.e., V/Cr and Th/U) has been shown to be an unreliable indicator of past seawater conditions (Algeo and Liu, 2020). I would advise that the authors use enrichment factors (Algeo and Liu, 2020) or trace-metal/aluminium normalization (Bennett and Canfield, 2020) instead. Furthermore, both U and V are readily incorporated into the crystal structure of apatite (Pan and Fleet, 2002) and this process continues during diagenesis. Using U- and V-based proxies in sedimentary phosphorites can lead to erroneous conclusions regarding the redox state of the formational environment (Lumiste et al., 2021).  

It was added to text

Additionally, the enrichment of both Mo (> 5 μg g−1/%) and V (> 23 μg g−1/%), with V not exceeding 46 μg g−1/%, provides compelling evidence for a euxinic basin-type depositional environment (Smith et al., 2010). Moreover, the enrichment of V (> 46 μg g−1/%), U (> 5 μg g−1/%), and Mo (> 5 μg g−1/%) serves as robust evidence for sediments depositing within the anoxic core of perennial oxygen-minimum zones environments (Jones et al., 2013). Conversely, the enrichment of U (> 1 μg g−1/%) coinciding with low enrichment of V (< 23 μg g−1/%) and Mo (< 5 μg g−1/%) is strong evidence of sediment deposition in the oxic water beneath the core of a perennial OMZ environment (Gupta and Das, 2017; Bennett and Canfield, 2020). Furthermore, both U and V are readily incorporated into the crystal structure of apatite (Pan and Fleet, 2002) and this process continues during diagenesis. Using U- and V-based proxies in sedimentary phosphorites can lead to erroneous conclusions regarding the redox state of the formational environment (Lumiste et al., 2021).

Lines 264-267 Suboxic conditions are defined by low amounts of dissolved oxygen (0.2-0.0 ml/1), and anoxic conditions are defined by the absence of oxygen (e.g., Tribovillard et al., 2006; Tyson and Pearson, 1991). Anoxic sediments can be further divided into sulfidic/euxinic conditions if there are free sulfides in the water column. Suboxic environments do not contain dissolved sulfides.

It was added to text

Table 5 Are these numbers concentrations? Perhaps these are PAAS-normalize values and not concentrations (ppm)?

Yes, These are REE concentrations of nodules, not PAAS-normalize values

Line 318-320 These are not true negative Ce anomalies, the negative Ce/Ce* values noted by the authors are likely caused by high Pr concentrations. I advise that the authors used the method proposed by Bau and Dulski, (1996) for classifying Ce-anomalies.

true, negative Ce anomaly in nodules may also be associated with low La and high Pr

Lines 318-344 I have serious issues with this discussion. Using REE concentrations in sedimentary phosphorites to reconstruct past seawater conditions is problematic. During precipitation, sedimentary apatite contains very low concentrations of REEs, and the vast majority of these elements are taken up from pore water during diagenesis.

Then, what is the source of high REE in sedimentary apatite?

Line 366-376 “It is suggested two mechanisms to provide phosphate ions from anoxic pore waters by Stumm and Leckie (1970).” What are the mechanisms? The formation of phosphorites is not described.

It was added to text

According to Stumm and Leckie (1970), there are two mechanisms to supply phosphate ions to anoxic pore waters; (1) The decomposition of phosphorus-containing organic materials and (2) the reduction of hydrous ferric oxides that bind phosphate to their surfaces under reducing conditions. 

Reviewer 3 Report

Comments included in the manuscript

Author Response

You can see all correction on manuscript

Reviewer 4 Report

In addition to my main comments, in a separate pdf, the authors should greatly expand the details of their methodological approach for analysing majors, traces, and REEs. The operational conditions of the instruments, whether or not the instruments can reproduce international standards, and ideally the actual per measurement uncertainties (these in a supplementary data file, to enable people to independently check data and easily download them to use in future work).

Author Response

Summary:

Sasmaz et al report novel measurements of the major element, trace, and REE compositions of phosphorite deposits of Ediacaran age. The authors stress that these phosphorites formed during an interval of great biogeochemical change; both with respect to the dominant macroscopic life forms and the prevailing redox conditions of aqueous environments. The key observation is that REE compositions have a normalised spider plot shape as well as some specific ratios that, whilst not unheard of, are less common, and may indicate some nuanced aspects of the initial near-surface formation environment of the phosphorites. In general, the paper appears to represent a solid contribution to the literature. However, there are several questions about the context of the new data that came to my mind whilst completing this review. I believe the authors can and should comment on these queries in more detail before the work is published. I have laid out these comments below, but in-short: the authors need to more fully contextualise and defend their findings with previous evidence, drawn from the literature. I recommend that the manuscript is potentially accepted after moderate revisions, but only if the key novel points of the work are given more context and detailed explanation/defence. I look forward to seeing the revised manuscript.

Authors: We thank reviewer and we appreciate her/his comment that “our paper appears to represent a solid contribution to the literature”. We will of course answer point by her/his comments below.

Major comments:

An aspect that is mostly missing from the discussion of the paper but would add greatly to the potential

impact in the field is: how exactly does this particular REE profile and high degree of enrichment fit into the

time-line of phosphorite deposits over time? The authors mention the work by Emsbo 2015 which has a

timeline for existing data. I would like to see the authors plot this again and add their own data. They should also make a second version of their own Figure 3 including more literature data. Finally, I would like to another similar figure to Fig. 3, but this time specifically plotting (Sm/Yb)n vs. Sample age for their own results and literature phosphorite data. These edits would, with some extra discussion, help to cover more thoroughly a few crucial points:

Authors: The reviewer is correct. The change has been done accordingly. Our data are plotted below. The figure shows consistent of our REE trend observed in the studied area (Fig. 7) and previous studies by Emsbo et al. (2015) and Yu et al. (2023). They propose that the formation of MREE-rich phosphates was influenced by variations in seawater composition across different time periods and regions. It was added to text in order to compare MREE trends of some phosphate deposits of Emsbo et al. (2015) and Yu et al. (2023).

-

Figure. Comparison of PAAS-normalized REE patterns in different sedimentary phosphates. Data from Emsbo et al. (2015) and Yu et al. (2023).

Fig. 3 currently seems to suggest that there are no similar phosphorite compositions to the present study. However, I have inserted here screenshots of similar looking phosphorites from Emsbo (2015) on the basis of the REE profile. In making a new plot of (Sm/Yb)n vs. age with a larger dataset that also includes these examples, do the authors find that these particular phosphorites plot in the same space as the deposit they have studied? i.e., do they have not just the same profile shape, but also the same specific ratios? If not, why is the studied deposit so rich in MREEs, specifically?  If similar, you think that these Silurian and Mississippian phosphorites (images above) formed via the same genetic model to the deposit you have studied?

Authors: It seems that REE trends vary locally over different geological time periods. For example, Ediacaran phosphates have a high REE content in our samples, while it is very low in Danzhai (Chinese) phosphates. This contrast of composition is likely linked the local conditions. For example, it’s widely observed that during High sea level period, we observe an increasing of bioproductivity leading to a high amount of REE and trace metal elements. Chemical composition of the ocean across geological time have been changed in term of concentration of REE and/or trace elements.  Ocean redox conditions and carbon cycles have varied significantly throughout geologic time (Jenkyns, 2010; Gill et al., 2011; McLaughlin et al., 2012; Harper et al., 2014).

If so, what model did previous studies propose for the above phosphorites, and is it consistent with your proposal? If not, why not? Either way, I would suspect that you can sell your results as evidence for a very ‘early’ occurrence of these specific phosphorite formation conditions – whatever they may be. By plotting a time-series of the relevant data, as I suggested above, you will be able to clearly make this point. An expanded discussion of the context of your findings, along the lines I mention here, will help people to understand the importance of your observations. The abstract should be modified to reflect any changes made of this kind.

Authors: Our model is similar to the previous study model. I agree with the reviewer that our phosphorite have been formed by early diagenetic reactions starting at the interface water/sediment.  

An aspect I was unclear on in the genetic model is why upwelling is needed? I would have guessed, since the precipitation of the P occurs in pore-waters, that all of the P comes from the organic matter (as the authors mention). So, is the upwelling simply fuelling primary productivity / life in the shallow water which, upon sinking, delivers the organics to the sediments which then undergoes oxidation and P-release to porewaters? If so, this should be reworded and expressed more clearly in the genetic model figure and description in the main text. 

Auhors: Upwelling is a great source of nutrients. This permit an increasing of bioproductivity and then accumulation of organic matter (OM). The biodegradation of this OM permit the P-realease which will be incorporated inside de nodules during diagenetic process.

Similarly, is initial biological matter actually REE-rich? Do the authors have references to prove this? Could the REEs instead come from settling iron oxide particles? The authors do mention iron oxides at one point, and I am aware of many iron oxide REE deposits elsewhere. I would suggest either justifying the current text or rethinking this subtle point if there is any reason to suspect that many REEs were initially delivered to the sediments by iron oxides, and then were partitioned into apatite during diagenesis. I am happy to be proven wrong, but – if so – I would like to see a more convincing case in the text.

Auhors: We have no evidence of presence of iron oxide. However we have clay layers and silica accumulation. Our hypothesis is that the REEs richness is probably linked the  combined sources including organic matter and clays.

Finally, can alteration of the apatite REE patterns/signals be ruled out? Is there any evidence of crosscutting veins, etc? It is important to clearly defend such points when the main interest relates to records of initial pore water chemistry, and I was not totally clear on whether or not this could be a concern in general and/or in this specific case from text as it is currently written.

Auhors: We agree with the reviewer that we cannot ruled out alteration of the apatite REE patterns/signals. However we have any evidence of apatite and/or crosscutting veins.

(see next page for minor comments)

Minor comments:

Fig. 6: are all nodules the same setting?

Authors: Yes they have

If not, can they be differentiated with a legend? (and something other than the default colour scheme, preferably) Fig. 5 is currently not very appealing to look at. It would look better at higher resolution. I suggest the authors may consider updating their figure making approach using free software, as this is a relatively easy way to speed up graph-making and result in high-looking final figures without too much effort. For example, the following script when used in jupyter notebook (a free python3- based tool – it should work on most/all computers, free to install) will gives the following plot when given a .csv input file of data. In this case it is partition coefficient data in high pressure experiments. The data column headers are ‘Pressure’ and ‘KdMg_Ca’).

.

The code that follows is intended to be pasted into e.g., jupyter notebook, with a few minor changes to be carried out before running. In yellow is my username on my computer. This text would need to be updated with relevant username for own computer. In blue is the file name for the data I plotted above as an example. This would also need changing. Highlighted in green are tools that you will need to install during python3 and jupyter notebook set-up before running this would work. In red is the set-up for the plot arrangement; you may want more or fewer components.

Authors: We are absolutely aware about the efficiency of this program. However, for some convenience we used Excel program

Round 2

Reviewer 2 Report

The readability and comprehensibility have significantly improved.  Some minor issues regarding some of the interpretations made by the authors are listed below.

However, I do have one significant issue with this manuscript – the genetic model provided by the authors. The main questions I have regarding the authors’ interpretations are: (i) Are the P and REE both delivered to the sediments by Fe-Mn oxides? (ii) Does the precipitation of the nodules and uptake of REE occur simultaneously or are the nodules formed first and REE are taken up during porewater-apatite interactions? (iii) If the primary source of REE is Fe-Mn particles, then how much valuable info can be gathered from these sediments regarding past seawater conditions? Fe-Mn oxides fractionate REE during their formation, and their REE patterns are not reflective of seawater conditions.

I would recommend that the authors describe their genetic model in more detail and address the minor comments listed below. I recommend moderate revisions to be made prior to publishing.

2.2. Minor comments:

Lines 46-47 This sentence is confusing. The cited article (Holland, 2005) states that between 0.8-0.54 Ga, BIF, Mn-deposits and phosphorite deposits reappeared in the sediment record.

Lines 67-68 a “well-ventilated” ocean is oxic, not anoxic.

Line 253-275 This paragraph needs to be rewritten. In its current form, the text has logical inconsistencies and it is hard to follow what the authors meant.

Line 272 This citation does not support these claims. Citation [71] is on REE+Y systematics, not V- and U-based proxies.   

Lines 341-343 Whereas positive Eu-anomalies are indicative of (extremely) reductive conditions, negative Eu anomalies do not form under oxic conditions. Under normal surface conditions (i.e., seawater or pore water) Eu/Eu* ratios are not significantly modified (Bau et al., 2010) and negative Eu-anomalies are likely caused by other factors (e.g., volcanic ash input; Zhao et al., 2013).

Lines 354-358 Or the negative Y-anomalies were inherited from Fe-Mn oxides that delivered the REE to the pore water?  

Lines 387-390 A third possible pathway for modifying pore water phosphate concentrations is polyphosphate hydrolysis by Large Sulfur Bacteria, mentioned in citation [4] (Schulz and Schulz, 2005).

Lines 397-399 Besides hydrogenous and hydrothermal sources, pore water diagenesis is an important source for REE in phosphates.  

References:

Bau, M., Balan, S., Schmidt, K., Koschinsky, A., 2010. Rare earth elements in mussel shells of the Mytilidae family as tracers for hidden and fossil high-temperature hydrothermal systems. Earth Planet. Sci. Lett. 299, 310–316. https://doi.org/10.1016/j.epsl.2010.09.011

Holland, H.D., 2005. Sedimentary mineral deposits and the evolution of Earth’s near-surface environments. Econ. Geol. 100, 1489–1509. https://doi.org/10.2113/gsecongeo.100.8.1489

Schulz, H., Schulz, H.N., 2005. Large Sulfur Bacteria and the Formation of Phosphorite. Science (80-. ). 307, 416–418. https://doi.org/10.1126/science.1103096

Zhao, L., Chen, Z.Q., Algeo, T.J., Chen, J., Chen, Y., Tong, J., Gao, S., Zhou, L., Hu, Z., Liu, Y., 2013. Rare-earth element patterns in conodont albid crowns: Evidence for massive inputs of volcanic ash during the latest Permian biocrisis? Glob. Planet. Change 105, 135–151. https://doi.org/10.1016/j.gloplacha.2012.09.001

Author Response

Responses to Reviewer 2

The readability and comprehensibility have significantly improved.  Some minor issues regarding some of the interpretations made by the authors are listed below.

However, I do have one significant issue with this manuscript – the genetic model provided by the authors. The main questions I have regarding the authors’ interpretations are: (i) Are the P and REE both delivered to the sediments by Fe-Mn oxides? (ii) Does the precipitation of the nodules and uptake of REE occur simultaneously or are the nodules formed first and REE are taken up during porewater-apatite interactions? (iii) If the primary source of REE is Fe-Mn particles, then how much valuable info can be gathered from these sediments regarding past seawater conditions? Fe-Mn oxides fractionate REE during their formation, and their REE patterns are not reflective of seawater conditions.

I would recommend that the authors describe their genetic model in more detail and address the minor comments listed below. I recommend moderate revisions to be made prior to publishing.

2.2. Minor comments:

Lines 46-47 This sentence is confusing. The cited article (Holland, 2005) states that between 0.8-0.54 Ga, BIF, Mn-deposits and phosphorite deposits reappeared in the sediment record.

It was corrected

Lines 67-68 a “well-ventilated” ocean is oxic, not anoxic.

It was corrected

Line 253-275 This paragraph needs to be rewritten. In its current form, the text has logical inconsistencies and it is hard to follow what the authors meant.

Contrasting Citations [66] and [71] were removed in this section.

Line 272 This citation does not support these claims. Citation [71] is on REE+Y systematics, not V- and U-based proxies.   

Citation [71] was removed

Lines 341-343 Whereas positive Eu-anomalies are indicative of (extremely) reductive conditions, negative Eu anomalies do not form under oxic conditions. Under normal surface conditions (i.e., seawater or pore water) Eu/Eu* ratios are not significantly modified (Bau et al., 2010) and negative Eu-anomalies are likely caused by other factors (e.g., volcanic ash input; Zhao et al., 2013).

This explanation was added to text

Lines 354-358 Or the negative Y-anomalies were inherited from Fe-Mn oxides that delivered the REE to the pore water?  

This sentence was added to text with Lumiste et al. [96]

Similarly, Chen et al. [95] reported that the ferromanganese oxides from 4071m water depth in the Gagua Ridge had negative Y anomalies. Lumiste et al. [96] indicated that the negative Y-anomalies were inherited from Fe-Mn oxides that delivered the REE to the pore water.

Lines 387-390 A third possible pathway for modifying pore water phosphate concentrations is polyphosphate hydrolysis by Large Sulfur Bacteria, mentioned in citation [4] (Schulz and Schulz, 2005).

It was added to text ‘’ and (3) a possible pathway for modifying pore water phosphate concentrations is polyphosphate hydrolysis by Large Sulfur Bacteria.’’

Lines 397-399 Besides hydrogenous and hydrothermal sources, pore water diagenesis is an important source for REE in phosphates.  

It was added to text:

Besides hydrogenous and hydrothermal sources, pore water diagenesis is an important source for REE in phosphates [107].  
